# Effect of Electrolysis on Activated Sludge during the Hydrolysis and Acidogenesis Stages in the Anaerobic Digestion of Poultry Manure

**Yelizaveta Chernysh** [1,2,3] ![ID], **Magdalena Balintova** [1,4,*] ![ID], **Vladimir Shtepa** [1], **Viktoriia Chubur** [1,2] and **Natalia Junakova** [4] ![ID]

1  International Innovation and Applied Center "Aquatic Artery", Sumy State University, 2 Rymskogo-Korsakova Street, 40007 Sumy, Ukraine; e.chernish@ssu.edu.ua (Y.C.); aquartery@ecolog.sumdu.edu.ua (V.S.); v.chubur@ecolog.sumdu.edu.ua (V.C.)
2  Department of Sustainable Technologies, Faculty of Tropical AgriSciences, Czech University of Life Sciences Prague, Kamýcká 129, 16500 Prague, Czech Republic
3  T.G. Masaryk Water Research Institute, Podbabská 2582/30, 16000 Prague, Czech Republic
4  Institute of Environmental Engineering, Faculty of Civil Engineering, Technical University of Kosice, Vysokoskolska 4, 04200 Kosice, Slovakia; natalia.junakova@tuke.sk
*  Correspondence: magdalena.balintova@tuke.sk; Tel.: +421-155-602-4127

**Abstract:** This paper focuses on the study of the effect of electrolysis on activated sludge in a microbial electrolysis cell (MEC) under the anaerobic digestion of poultry manure. This study was conducted using a bioreactor design with and without electrodes (conventional condition). Measurements of pH, redox potential (ORP), and total dissolved solids were carried out, as was the microscopy of activated sludge during treatment and gasometry. There was an increase in the yields of $CH_4$ and $CO_2$ compared to conventional conditions. Thus, on the 14th day, there was an increase in the $CH_4$ yield to 35.1% compared with the conventional conditions—31.6%—as well as in the $CO_2$ yield to 53.5% compared with the cell without electrodes—37.7%. Visually, the microscopy of anaerobic activated sludge showed changes in the aggregation process itself, with the formation of cells of clusters of microorganism colonies with branches of a delineated shape. ORP fluctuations were related to the process of the dissociation into ions during the passage of an electric current through the electrodes, and were observed before and after the inclusion of a current into the system. A model of the effect of electrolysis during anaerobic digestion was developed, taking into account the influencing factors on the condition of the activated sludge.

**Keywords:** anaerobic digestion; electrolysis; microbial electrolysis cell; activated sludge; poultry manure





## 1. Introduction

Electrolysis treatment processes are more frequently the subject of research in the field of managing organic waste, sewage sludge, and activated sludge in combination with bioprocesses [1]. Therefore, to improve the anaerobic digestion of wastewater in the study by Barrios et al. (2021) [2], an electro-oxidative pretreatment was carried out with various concentrations of total solids (TSs). A larger reduction in chemical oxygen demand (COD) and volatile solids was achieved with a sludge pretreatment at a current density of 21.4 mA/cm$^2$ and 3.0% TSs. The increase in biogas production indicates that the maximum degradation and methane production are directly related to the applied current density. Most of the organic content of the activated sewage sludge consists of hard-to-degrade microbial cells, which have to be pretreated to produce energy through anaerobic digestion; therefore, the electro-oxidative pretreatment was proven to be effective. It should be noted that this study used a boron-doped diamond electrode to increase the rate of hydrolysis by destroying the cell walls of the activated sludge [2]. The electrode material is expensive, so making the process cheaper and solving several other technical problems remain open. It

should be noted that there is a limited number of works using an electrolysis treatment to improve biogas production in the statistical data obtained in the Scopus and Web of Science databases (Figure 1). Nevertheless, a Google Scholar search shows 16,400 views from 2010 to 2022, with a persistent increase in the number of papers, including 1910 published studies from 2022 (Figure 1).

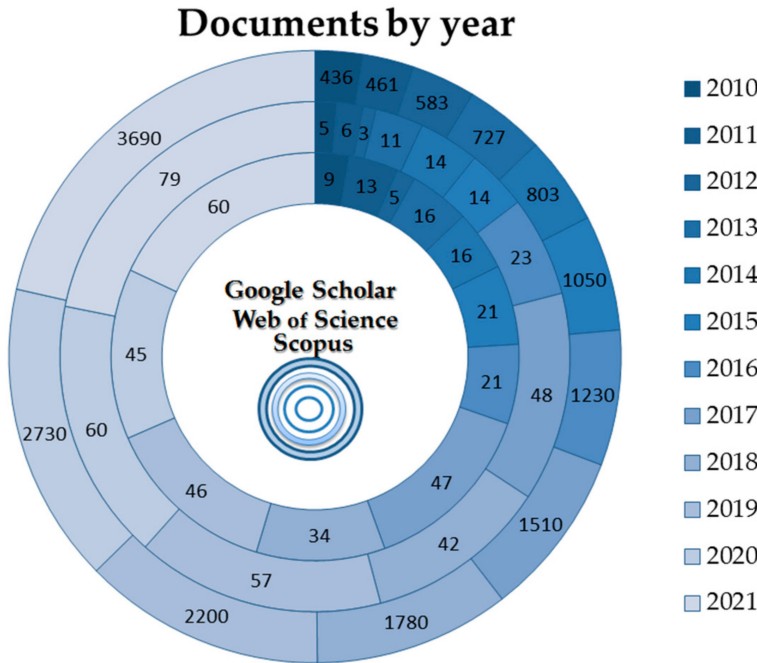

**Figure 1.** Publication activity, keywords "biogas" and "electrolysis": Scopus database; Web of Science database; and Google Scholar.

Thus, the interest in this area of research is increasing rapidly. Currently, there is a growing number of works related to the study of the impact of electrolysis on anaerobic digestion, the peculiarities of activated sludge transformation under such an influence, and biogas production.

Research trends are represented in various areas of research, including wastewater treatment with increased methane production in anaerobic reactors equipped with electrodes (Tartakovsky et al. (2011), Rani et al. (2022) [3,4]); the improvement of design features of unit combinations to produce more hydrogen and methane (Rader et al. (2010), Bo et al. (2014) [5,6]); the influence of inoculum origin on the development of microbial associations in microbial electrolysis cell bioreactors (Cerrillo et al. (2017) [7]); and microbial associations that determine the biochemical parameters of anaerobic waste processing to obtain biofuel (Shulipa et al. (2017) [8]).

In previous studies on poultry litter treatment by Wang X. et al. (2015) and Wang M. et al. (2016) [9,10], a combination of anaerobic digestion followed by electrolysis was used to pretreat the anaerobically digested effluent and then used as an algae growth medium.

In the paper by Wang et al. (2021) [11], anaerobic digestion combined with an electrolysis cell accelerates methane production from hydrolysate biomass. Authors investigated the fact that methane productivity increased by 7.8 times (raw activated sludge) and 2.1 times (pretreated activated sludge) at an applied voltage of 0.8 V compared to productivity without applying voltage. The applied voltage with raw activated sludge enriched the electricigens as well as methanogens and purposely enriched the fermentative bacteria as well as syntrophic acetogenic bacteria in both of the electrode biofilms. Based on the conclusions of [11], the boosted hydrolysis–fermentation and synergy of acetogenic bacteria and hydrogenotrophic methanogens may be the main reason for the continued high efficiency of methanogenesis.

The study by Heng et al. (2021) [12] showed that an electrochemical pretreatment could improve the disintegration and dewatering of activated sludge by increasing the soluble organic fraction or sCOD, and at the same time reducing the concentration of solids and the capillary suction time. Moreover, it has been proven that integrating an optimally working electrochemical–anaerobic digestion system as an activated sludge pretreatment strategy can increase the production rate and biogas capacity by 44–67% compared to a conventional process using only anaerobic digestion [12].

Yu et al. (2021) [13] also proved that microbial electrolysis cells (MECs) proved the enhanced degradation of complex organic matter to produce hydrogen from waste sludge lysate. The microbial electrolysis cell increased the removal efficiency of chemical oxygen demand by 40.33% and achieved an average hydrogen production rate 79 times higher than the anaerobic fermentation system [13].

The aspect of the effective regulation of electron flow was considered by Gao et al. (2021) [14]; the possibility of accelerating biodegradation and methanogenesis through microbial electrolysis compared to regular anaerobic digestion. However, the practical application and scale-up require a deep understanding and further exploration of the size and distribution of the electrodes. Sufficient contact time, well flow field, and an appropriate electrode surface area are three efficient methods to improve organic removal efficiency and increase methane production, as well as reactor design purposes. Reactor types with different cathode space ratios were investigated to develop a well flow pattern to produce high methane production. Considering organic removal, methane recovery, electron generation, and material consumption, the recommended cathode space ratio according to [14] is 1.33 $cm^2/cm^3$. The work also noted an enrichment of the dominant functional microbes involved in organic removal, methane reduction, and electron generation, including *Enterococcus, Desulfovibrio, Methanosarcina, Methanobacterium, Methanospirillum, Methanobrevibacter*, and *Methanocorpusculum*.

The results of Li et al. (2021) [15] show that a significant increase in the production of volatile fatty acids can be achieved by electrically assisted acidogenic fermentation as well as by secondary acidogenic fermentation through the application of voltage. The application of voltage is considered to have the effect of resolubilization and hydrolysis on sludge organics by loosening the sludge structure (changing the values of the fractal size and the associated functional groups) and stimulating the activity of particular enzymes.

A combined system of an MEC and anaerobic digestion was used to increase methane production from the activated sludge of wastes, according to Bao et al. (2021) [16]. The combined microbial electrolysis cell and anaerobic digestion system used alkali, ultrasound, and alkaline as well as high-temperature microaeration as pretreatment methods to disintegrate the flocculus of sewage sludge and destroy bacterial cells. The results showed that microorganisms oxidized organic matter at the anode, and that hydrogen was synthesized at the cathode. Hydrogen was then used as a substrate for hydrogenotrophic methanogens to stimulate methane production. Consequently, compared to a standard bioreactor, a combined microbial electrolysis cell and anaerobic digestion system has more advantages in the energy conversion of activated sewage sludge. The addition of a bioelectrochemical system can promote methane production.

In our previous study by Shtepa et al., 2021 [17], we also substantiated the relationship between the effects of advanced oxidation technologies on the efficiency of an activated sludge microbial association. It should be noted that the electrochemical anaerobic digestion system provides better effluent and increases biogas production. It has the potential to optimize important process variables that can increase sludge degradation and biogas production. In addition, response surface methodology models can be used in the process automation and management of wastewater treatment systems for more feasible applications in the industrial process, which require further study.

MECs are a promising area for the production of hydrogen from organic matter, including sewage sludge and other organic waste. The possibility of using an MEC for biomethane production is also of interest. However, a close search of three databases

(Scopus, Web of Science, and Google Scholar) on the electrolysis treatment of quail manure during anaerobic digestion showed a gap in the study of this substrate. Likewise, there are only a few specific studies on quail manure.

A case study by Onursal et al. (2021) [18] determined that quail manure has a relatively high biogas potential. As a result of a separate study, the cumulative biogas and methane production of quail manure was higher than cow and rose oil processing waste.

Quail manure was the subject of research carried out by Cerrillo et al. (2017) [19]; it was determined to be high in total solids, with three–four volatiles, as well as high in nitrogen and potassium. Comparing the methane potential of chicken and quail litter with additional supplements, the chicken substrate has a higher proportion of methane and a higher total biogas yield (Jijai et al. (2020), Silva et al. (2021) [20,21]) than the quail substrate due to the C/N ratio, which can cause ammonification in the reaction medium; the lowest C/N ratio was of quail litter, indicating a high level of nitrogen due to its feed.

Thus, this study aims to investigate the effect of electrolysis on activated sludge in an MEC during the anaerobic digestion of poultry manure. To achieve the aim of the study, the following tasks were carried out:

- Microscopy of anaerobic activated sludge during anaerobic digestion in MECs and cells without electrodes for comparison;
- Study of the qualitative and quantitative compositions of the obtained biogas under conventional conditions (cell without electrodes) and with electrolysis (two-chamber MEC), with a focus on the formation of methane in the biogas;
- Study of changes in the values of pH, ORP, and TDS under conventional conditions (cell without electrodes) and with electrolysis (two-chamber MEC);
- Formalization of a generalized model of the effect of electrolysis on the anaerobic digestion process based on the data obtained and previous studies.

## 2. Materials and Methods

### 2.1. Characteristics of Substrate—Poultry Manure

Quail manure from a household farm was used as a substrate. As a gift, 25 kg of manure from a local farm, called "Zlak" (Sumy region, Ukraine), was provided.

Quail manure contains high amounts of nitrogen compounds and has a characteristic ammonia odor. The basic elemental composition and characteristics of poultry manure are presented in Table 1.

**Table 1.** Physicochemical composition of household quail manure.

| Parameter | Measurement |
|---|---|
| Appearance, color, and odor | Dark brown, crumbly mass, with a specific odor, and with sand inclusions |
| Mass fraction of water, % | 50–65 |
| Mass fraction of macro elements, %: | |
| Nitrogen | 0.7 |
| Phosphorus | 0.5 |
| Potassium | 0.85 |
| Acidity, pH | 7.5–8.0 |

The moisture content of the raw manure is 71.21%. The manure is poorly soluble in water. Separation with the sedimentation of the solid phase occurs in 10–15 min (Figure 2).

The quail manure substrate pretreatment was conducted by diluting 400 g of fresh manure in 2600 mL of tap water and incubated at a temperature of 35 °C for 48 h under anaerobic conditions without light. The obtained parameters for the substrate were as follows: pH, 9.03; redox potential (ORP), −425 mV; and total dissolved solids (TDS), 604 ppm. In tap water, the amount of chloride was 221 mg/L and pH = 6.7.

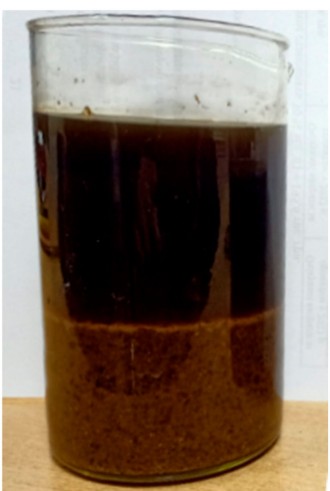

**Figure 2.** General view of the substrate, diluted 1/6 with tap water.

*2.2. Characteristics of Innoculate—Activated Sludge*

Activated sewage sludge from anaerobic transformation systems of agroholding "Orel Leader" (Dnepropetrovsk region, Ukraine) was used. The inoculum was stored in a refrigerator at a constant temperature of 4 °C. The inoculum was thawed and reconditioned for the experiments. The pretreatment of activated sewage sludge required incubation at a temperature of 35 °C for 48 h under anaerobic conditions without light. The obtained parameters for the inoculum were as follows: pH, 6.90; ORP, −390 mV; and TDS, 112 ppm.

*2.3. Laboratory Set-Ups*

2.3.1. Two-Chamber Membrane Bioreactor MEC Used in the Study

Two-chamber MECs (Figure 3) were made in the laboratory using a thermoplastic (polypropylene), with a total volume of 6 dm$^3$ (15 × 13 × 30 cm). The anode (2.5 × 2.5 × 20 cm) and cathode (2.5 × 2.5 × 20 cm) were made of graphite and placed in reactors separated by a membrane in the proportions of 2/3 of the tank volume–cathode part and 1/3 of the tank volume–anode part. The distance between the anode and cathode was 10 cm. The cells were equipped with a sampling point for gas and sludge sampling. A cell without electrodes was used as a control.

The electrolysis treatment was carried out by connecting a power supply to the electrodes, applying a current of 20 A for 5 min every day. This treatment mode allowed us to start work with stable biogas generation with a hydrogen content of 18.9–20.8%. The hydrogen content was measured by a chromatographic method using a thermal conductivity detector using a laboratory gas chromatograph: SELMICHROM-1 (Enterprise Selmy, Ukraine).

2.3.2. Control Parameters of the Anaerobic Digestion Process

The parameters that were measured are presented in the flowchart in Figure 4.

In this study, the state of the activated sludge and the process of biogas production during 12–14 days of anaerobic conversion were evaluated. The pretreatment included the stabilization of the inoculum and poultry manure at 35 °C in hermetically sealed containers for 48 h. The hydrolysis and acidogenesis of activated sludge were observed in a periodic mode to evaluate the effect of the electrolysis treatment on parameters such as pH, ORP, TDS, and biogas composition at these stages of anaerobic digestion, with an analysis of the overall condition of activated sludge by microscopy.

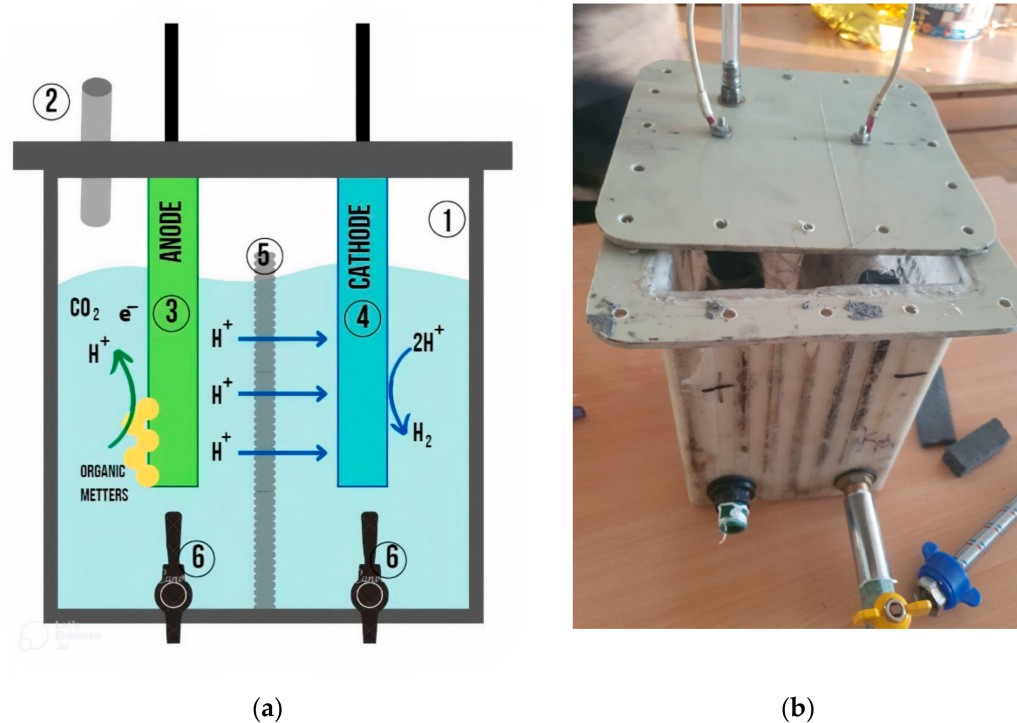

**Figure 3.** The laboratory set-up: (**a**) scheme; (**b**) photo: 1—double-chamber membrane bioreactor body; 2—branch pipe for gas phase extraction; 3—cathode; 4—anode; 5—semipermeable membrane made of polymer material; and 6—sampling holes from the cathode and anode areas of the bioreactor.

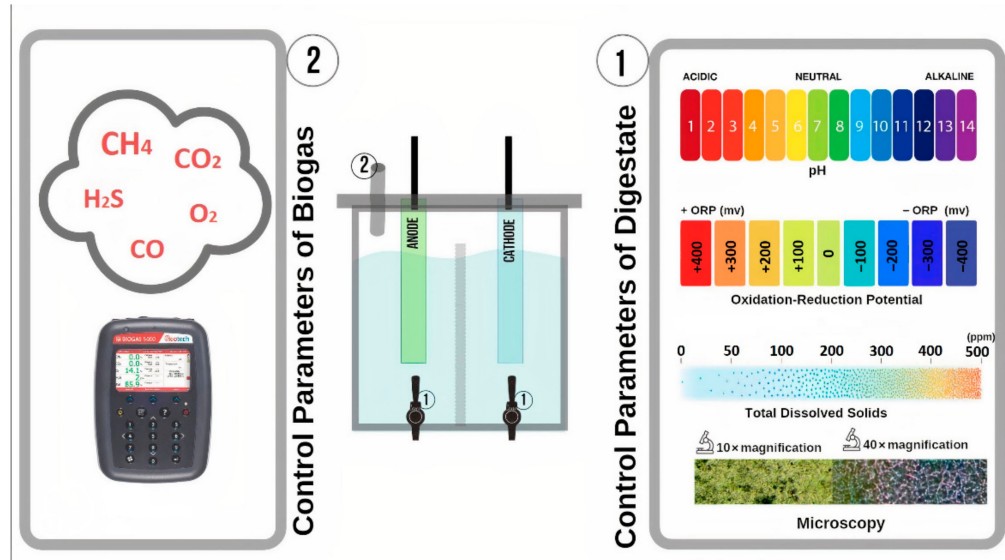

**Figure 4.** Control parameters of the anaerobic digestion: 1—liquid sampling; 2—gas sampling.

The moisture is determined by drying peat or an organic sample at $110 \pm 5\ ^\circ$C (ASTM D2974-14 Standard Test Methods for Moisture, Ash, and Organic Matter of Peat and Other Organic Soils).

To measure the hydrogen potential, a portable pH meter with an ATC function and backlight display was used; a PH-911 (Kelilong Electron, China) pH meter with the following characteristics: electrode sensitivity, $-0.01$ pH; measurement accuracy, $-0.01$ pH; and measurement error, $\pm 0.1$ pH.

To control the redox potential, an ORP-2069 (Shen Zhen Yage Technology Co., Ltd., Shenzhen, China) ORP meter was used with the following characteristics: measuring range of $-1999$ mV to 1999 mV; accuracy of $\pm 5$ mV.

A TDS-3 (HM Digital, China) portable total dissolved solids detector was used with the following characteristics: measuring range of liquids' hardness, 0–9990 ppm (mg/L); a precision of $+/-2\%$.

A Geotech BIOGAS 5000 (Cadmus Products, United Kingdom) multigas analyzer was used for measuring the component composition of biogas. This analyzer provides the precise control of gases, such as $CH_4$, $CO_2$, CO, $O_2$, and $H_2S$. The technical specifications are presented in Table 2.

**Table 2.** Accuracy of biogas components' measurements.

| Gas Cells | Range | Typical Accuracy (Range: Accuracy) | Typical Accuracy (Range: Accuracy) |
|---|---|---|---|
| $CH_4$ | 0–100% | 0–70%: $\pm 0.5\%$ (vol) | 70–100%: $\pm 1.5\%$ (vol) |
| $CO_2$ | 0–100% | 0–60%: $\pm 0.5\%$ (vol) | 60–100%: $\pm 1.5\%$ (vol) |
| $O_2$ | 0–25% | 0–25%: $\pm 1.0\%$ (vol) | |
| $H_2S$ | 0–5000 ppm | $\pm 2.0\%$ FS * | |
| CO | 0–2000 ppm | $\pm 2.0\%$FS * | |

* Full-scale.

The digestate samples were obtained after drying the liquid phase at 105 °C. The determination of the forms and cell structure was conducted on an EMV 100 AK transmission electron microscope (IEEE Electron, Sumy, Ukraine). The samples were also studied using a Binocular Biological Microscope XS-5520 LED MICROme (LLC "Trading House "MICROMED", Poltava, Ukraine). Microphotographs of microbial preparations were obtained and processed using a digital image output system, SEO Scan ICX 285 AK-F IEE-1394, and morphometric software, SEO Image Lab 2.0 (Sumy, Ukraine); images of microbial colonies were obtained using a Canon Power IS digital camera with 10 MPx matrix separation.

The gas phase was collected in a 5 L sampling gas bag (Casta Vinodelov, Ukraine) with a shutoff valve. The gas volume was determined by the water column displacement method. For the measurements, the exact volume of water in the reservoir was measured by taking the empty gas bag into account. During the measurement period, the gas bag connected to the fermenter was shut off by a valve and put into the reservoir with an exact volume of water. The difference in the water displaced is the volume of gas created during a certain period.

## 3. Results and Discussion

### 3.1. Microscopy of Anaerobic Activated Sludge during Anaerobic Digestion in MECs and Cells without Electrodes for Comparison

Active sludge flakes (Figures 5a and 6a) are formed as a result of bacterial bioflocculation. The stirred substrate sludge flakes are dynamic structures: large flakes are destroyed in the turbulent liquid flow, and small flakes clump into larger ones when they collide. Typically, the size of flakes is around 10,200 μm, but with slight agitation the flakes quickly enlarge, reaching a size of 1.5 mm. Large living protozoa (bristle worms, rotifers) were not detected.

Activated sludge flocs are multicellular systems combined into aggregate formations due to the binding of biologically originated polymeric gel produced by bioflocculation-producing bacteria, which potentially represent 90% to 95% of the total biomass. The remaining biomass is occupied mainly by protozoa and higher-level multicellular microorganisms. These bacteria feed on flocculogenic bacteria, thereby stimulating their physiological activity, which also affects the artificial biocenosis's purifying abilities. An important aspect in activating the development of anaerobic activated sludge is the ratio of substrates and inoculum, and several studies are devoted to it [22–26].

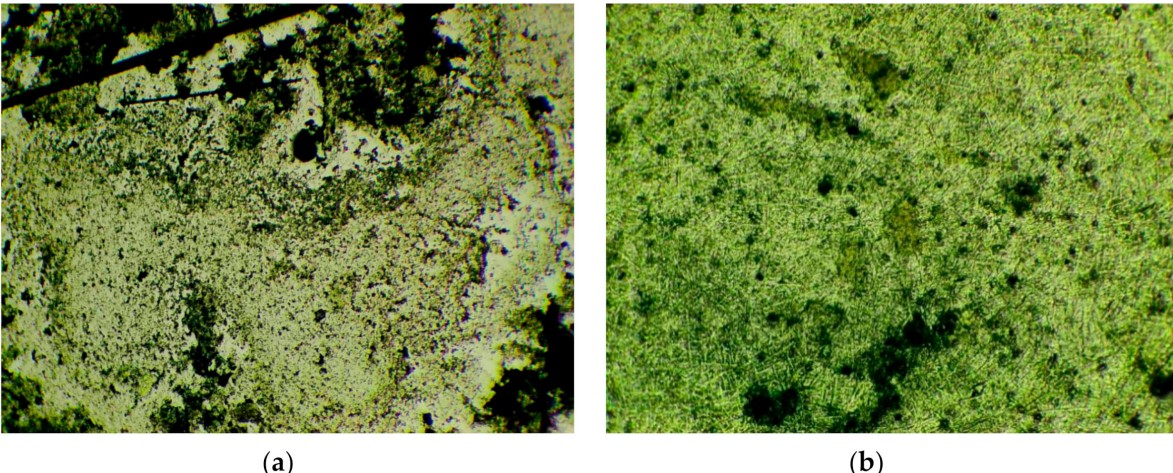

(**a**)　　　　　　　　　　　　　　　　　(**b**)

**Figure 5.** Anaerobic activated sludge in the process of poultry manure digestion, at $10\times$ magnification, light microscopy: (**a**) without electrolysis treatment; (**b**) with electrolysis treatment for 5 min twice a day.

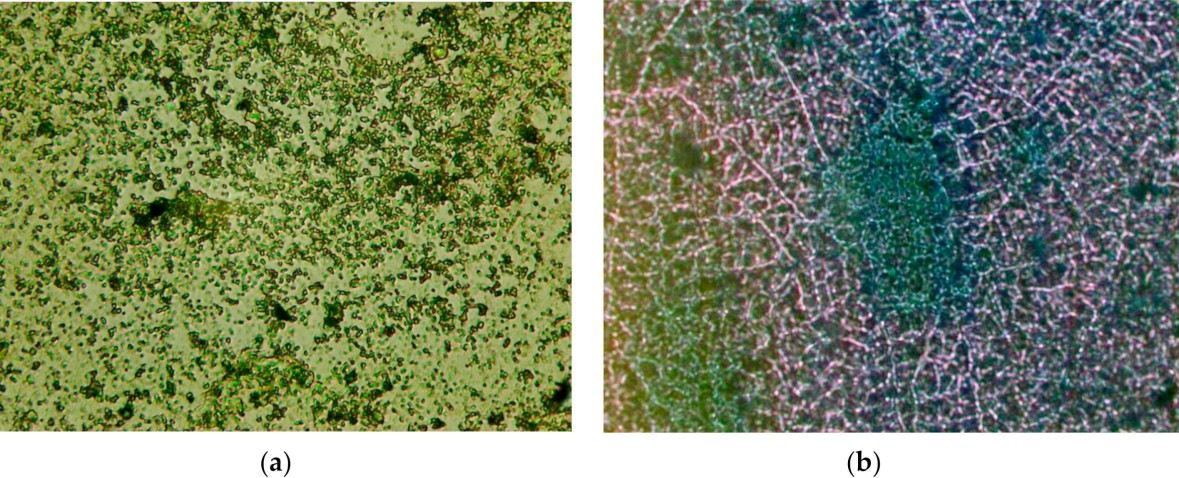

(**a**)　　　　　　　　　　　　　　　　　(**b**)

**Figure 6.** Anaerobic activated sludge in the process of poultry manure digestion, at $40\times$ magnification, light microscopy: (**a**) without electrolysis treatment; (**b**) with electrolysis treatment for 5 min twice a day.

Thus, the study by Gaur and Suthar (2017) [27] showed that the type and combination of inoculum with food waste are the main factors that can improve the codigestion process. The combination of cow dung, acclimatized anaerobic granular sludge, and activated sludge with food waste (1:1:1:1:1) proved to be the best codegradation inoculum to improve the biomethane production process. The experimental results agreed well with the Gompertz and first-order models for all of the waste combinations.

It is important in our research to test and optimize the mode of operation using the combination of an anaerobic bioreactor with an electrolysis cell; therefore, we followed one option of combining the substrate (poultry manure) with the inoculum. Further studies will be continued to study the effects of cosubstrate ratios as well.

As can be seen from Figures 5b and 6b, the structure of anaerobic sludge changes in comparison with initially activated sludge; under the influence of an electric current, toxic or nondegradable substances are oxidized to being biodegradable or completely oxidized to $CO_2$ and $H_2O$.

Thus, during the microscopy of anaerobic activated sludge after exposure to the electrolysis treatment, there was a change in the aggregation process, with the formation of clusters of microbial cell colonies with clearly delineated branches (Figure 6b). This result is related to the purpose of the research: to study the effects of electrolysis in anaerobic

digestion, using poultry manure with activated sludge inoculum from municipal sewage treatment plants as an example. The determination of changes in the structure of the activated sludge confirms the qualitative changes in the inoculum as one of the effects of the treatment process.

*3.2. Study of the Qualitative and Quantitative Composition of the Obtained Biogas under Conventional Conditions and with Electrolysis (Two-Chamber MEC)*

The bioprocess is carried out in the usual way, and is activated by an electric discharge between the electrodes in the two-chamber MEC. Active gas production began on the sixth day. Without electrolysis, the quail droppings were digested slower, and the biogas yield was insignificant on the sixth and eighth days, appearing to be 12 mL and 18 mL, respectively, but on the sixth and eighth days, the MEC produced 800 mL and 825 mL of biogas, respectively. On the 10th day, the biogas yield stabilized at 250 mL without treatment and 812 mL in the MEC. After that, the amount of biogas under standard conditions without treatment began to grow rapidly and, on the 14th day, reached 850 mL, while in the MEC the level of growth remained the same (Table 3). There was a significant increase in the yield of biogas with treatment, possibly due to the specific effect of the electric current on complexly degradable quail manure. These findings are supported by previous studies conducted on substrates such as agroindustrial wastewater [28], municipal sewage sludge [29,30], and liquid waste from the wine industry [25,31].

**Table 3.** Biogas yield volume under conventional conditions (cell without electrodes) and with electrolysis (two-chamber MEC).

| Parameter | Two-Chamber MEC | | | | | Cell without Electrodes | | | | |
|---|---|---|---|---|---|---|---|---|---|---|
| | Day 6 | Day 8 | Day 10 | Day 12 | Day 14 | Day 6 | Day 8 | Day 10 | Day 12 | Day 14 |
| V biogas, mL | 800 | 825 | 812 | 810 | 821 | 12 | 18 | 25 | 150 | 850 |

On the sixth day of treatment, the $CO_2$ yield increased (24.0% in the conventional conditions and 27.3% in a two-chamber MEC) and the $CH_4$ yield increased (4.7% in the conventional conditions and 8.3% in a two-chamber MEC). On the eighth day there was an increase in the $CH_4$ yield to 15.3% compared with the conventional treatment (control experiment)—11.5%—as well as in the $CO_2$ yield to 45.2% compared with the conventional treatment—41.0%. On the 14th day there was an increase in the $CH_4$ yield to 35.1% compared with the conventional treatment—31.6%—as well as in the $CO_2$ yield to 53.5% compared with the conventional treatment—37.7% (Table 4).

**Table 4.** The qualitative composition of obtained biogas under conventional conditions (cell without electrodes) and with electrolysis (two-chamber MEC).

| Gas | Parameter | Two-Chamber MEC | | | | | Cell without Electrodes | | | | |
|---|---|---|---|---|---|---|---|---|---|---|---|
| | | Day 6 | Day 8 | Day 10 | Day 12 | Day 14 | Day 6 | Day 8 | Day 10 | Day 12 | Day 14 |
| *Quality* | $CH_4$, % | 8.30 | 15.30 | 27.20 | 32.30 | 35.10 | 4.70 | 11.50 | 24.30 | 25.10 | 31.60 |
| | $CO_2$, % | 27.30 | 45.20 | 49.90 | 50.10 | 53.50 | 24.00 | 41.00 | 40.20 | 39.60 | 37.70 |
| | $O_2$, % | 1.80 | 1.50 | 1.30 | 1.10 | 0.50 | 3.40 | 2.40 | 2.10 | 1.80 | 1.50 |
| | CO, ppm | 5000+> | 5000+> | 565 | 431 | 236 | 5000+> | 5000+> | 5000+> | 5000+> | 5000+> |
| | $H_2S$, ppm | 5000+> | 3382 | 1251 | 693 | 487 | 5000+> | 5000+> | 5000+> | 5000+> | 5000+> |

The graph shown in Figure 7 demonstrates a comparison of the volumes of the methane yields in biogas.

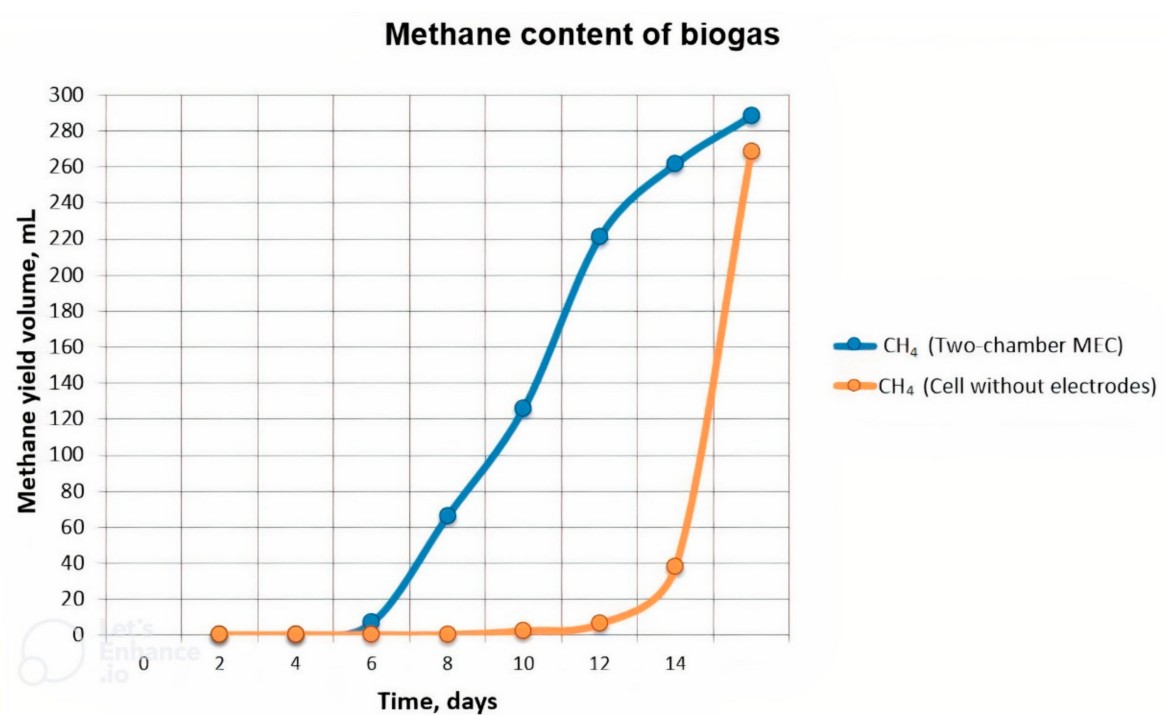

**Figure 7.** Comparison of the changes in CH$_4$ volumes in experiments with MECs and conventional conditions.

Therefore, a significantly higher biogas yield was achieved in a shorter treatment time, corresponding with Dalkilic's and Ugurlu's (2016) study, where the selection of a hydraulic retention time for an anaerobic digester with a microbial electrolysis cell system was taken up from 5 days to indicate that the system can achieve a similar treatment efficiency as digesters that process cattle manure with anaerobic digestion with a relatively long hydraulic retention time (10–30 days) [32].

Consequently, the electric discharge, affecting the growth of methane-forming bacteria, makes it possible to obtain energy due to the reduction of CO$_2$ to methane by the following reaction:

$$CO_2 + 4H_2 \rightarrow CH_4 + 2H_2O \tag{1}$$

Biogenic hydrogen is significantly demanded as an electron donor not only for the development of methanogenic archaea but also by sulfate-reducing bacteria. Accordingly, additional exogenous hydrogen is introduced via electrolysis, which activates the growth of lithotrophs as well as methane bioproduction.

Hydrogen is significantly demanded as an electron donor not only for the development of methanogenic archaea but also by sulfate-reducing bacteria. Accordingly, additional exogenous hydrogen is introduced via electrolysis, which activates the growth of lithotrophs as well as methane bioproduction.

It should be noted that the hydrogen sulfide content of the biogas from the MEC is considerably lower than that of the conventional treatment. Thus, in the MEC initially, the hydrogen sulfide concentration was more than 5000 ppm, as in conventional conditions. During this treatment, it fell on the 14th day to 487 ppm. At the same time, in the conventional treatment (cell without electrodes), the hydrogen sulfide concentration in all of the periods exceeded 5000 ppm (Table 4). Several previous studies have found that electrolysis reduces toxic impurities in biogas from the anaerobic digestion of wastewater and organic waste [3,33].

While considering the complexity of the biodegradable substrate, a strong oxidation pretreatment seems promising for improving biogas production and stimulating microbial growth, i.e., enhancing methanogenesis, which agrees with the results of studies conducted

on another type of substrate in [31], and the anaerobic digestion of liquid waste from the wine industry, with a different design and regime of electro-oxidative treatment.

### 3.3. Study of Changes in Values of pH, ORP, and TDS in MECs and Cells without Electrodes for Comparison

At the stage of hydrolytic degradation under facultatively anaerobic conditions, the $CO_2$ yield increased, and there was an initial shift to the alkaline side of pH as well as further stabilization on days four to six at a neutral level, which was similar to both under standard conditions and during electrolytic treatment anaerobic fermentation. In the pH range of 6.0 to 7.5, anaerobic processes are controlled by the interaction of the carbonic acid system and the net strong base, which was studied in [34,35]. From the 8th to 10th days, the pH stabilizes at 7.5–8.0 (Figure 8) and the TDS ranges from 455 to 535 ppm, indicating increased mineralization of the substrate during digestion.

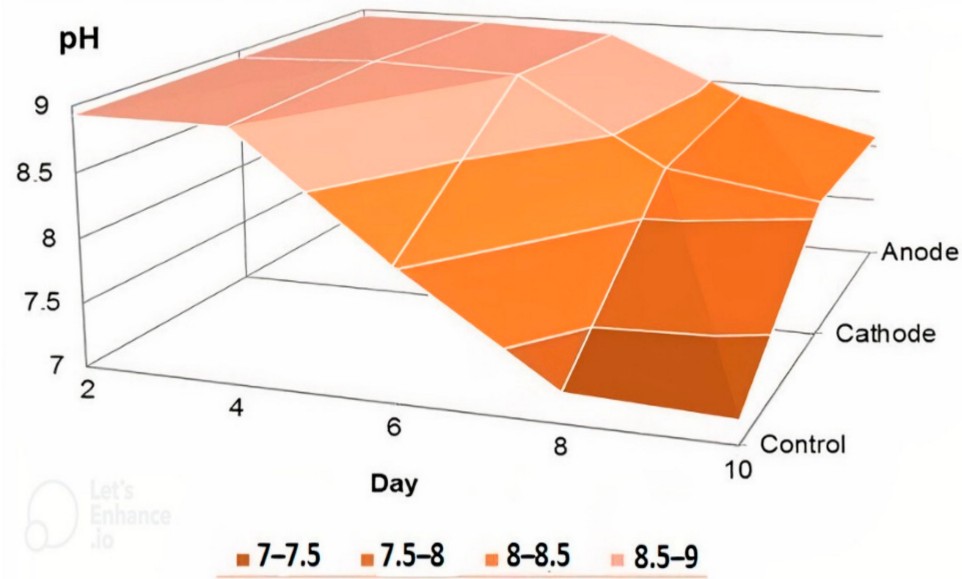

**Figure 8.** Comparison of changes in pH values in experiments with MECs.

The ORP in this case has a fluctuating character, with a decrease to −450 mV and an increase to −50 mV (Figure 9). Long D. Nghiem et al. (2014) evaluated the use of the ORP in controlling the injection of a small amount of oxygen into an anaerobic bioreactor to reduce the $H_2S$ concentration in the biogas. The results confirm that a micro-oxygen injection can effectively control $H_2S$ formation without impairing performance [36].

After 10 days there is a stabilization of the value in the low ORP levels, which corresponds to the state of strict anaerobiosis and is an indicator of the transition to the next stage of anaerobic digestion. Such ORP fluctuations were related to the process of the dissociation into ions during the passage of an electric current through the electrodes, and were observed before and after the inclusion of a current into the system. The decrease in the ORP was due to the metabolic activity of microorganisms. It is noteworthy that the correlation between pH values, the ORP, and the metabolic activity of methanogenic associations has been studied quite deeply in the processes of anaerobic digestion [37], and is following previous studies on the effect of an electric current on the fermentation of various substrates [38].

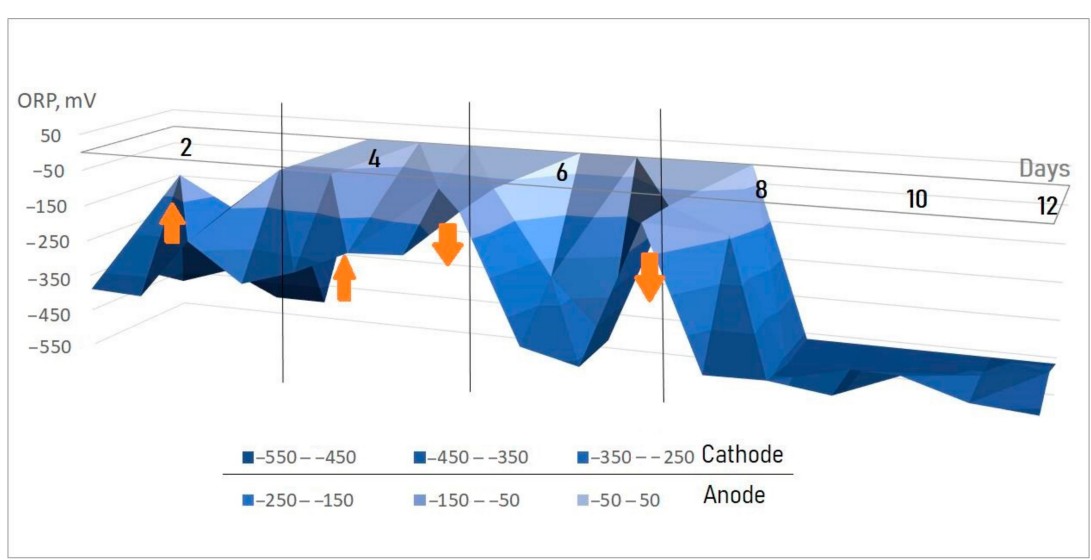

**Figure 9.** Comparison of changes in redox potential (ORP) values in experiments with MECs.

*3.4. Formalization of a Generalized Model of the Effect of Electrolysis on the Anaerobic Digestion Process Based on the Data Obtained and Previous Studies*

In this study, we were interested in the hydrolysis and acidogenesis stages, so the conversion period was taken appropriately as the basis for studying the stimulating effects of exposure to electrolysis precisely as a precursor to dark fermentation and/or methanogenesis. In a study by Jafari and Botte (2021) [39], where an electrochemical method in an alkaline environment was studied to break down a sludge structure at room temperature, a reduction of 24.85% of total solids and 46.42% of volatile solids was achieved, which reduces sludge disposal costs by about 25% compared to conventional treatment methods. The post-treatment sample characteristics showed that a large amount of organic material was released from the sludge samples into the liquid phase, indicating the potential to reduce the residence time in anaerobic digesters and achieve higher biogas production rates. The applied treatment demonstrated the possibility of removing pathogens and producing biosolid sludge for safe disposal in landfills or application in agriculture as a fertilizer [39]. It confirms our electrolytic treatment results during the initial hydrolytic phase of the anaerobic digestion of poultry manure solution and the active sludge of wastewater treatment facilities in a ratio of 2:1.

It should be noted that the use of electrolysis to intensify dark fermentation has several confirmations of the positive effect. For example, Marone et al. (2017) studied the production of biohydrogen from agroindustrial wastewater and byproducts by combining dark fermentation and microbial electrolysis in a two-step process. The total hydrogen production by combining dark fermentation and microbial electrolysis was increased 13-fold compared to fermentation alone, achieving a maximum total hydrogen yield of $1608.6 \pm 266.2$ mLH$_2$/gCOD and a maximum COD removal of $78.5 \pm 5.7\%$ [28]. These data are consistent with those obtained by us, but we used a different design solution, implementing the process in single-space bioreactor electrolysis and fermentation. Moreover, the possibility of using dark fermentation and a combination of microbial electrolysis cells with the intensification of hydrogen yield was substantiated even earlier in Kuppam et al. (2017) [40]. Considering hydrogen's relevant and renewable characteristics has led to the improvement of various biological processes for hydrogen production. Nevertheless, the commercialization of the biological process depends on improving the process design along with understanding the nature of hydrogen-producing communities and optimizing the process.

Considering the biodegradable substrate's complexity, strong oxidation pretreatment seems promising to improve biogas production and stimulate microbial growth at the

terminal stage, i.e., to enhance methanogenesis, which is also compatible with the research in [31]. It is increasingly relevant to prevent the negative impact of emerging contaminants, such as pharmaceuticals, particularly antibiotics, on the anaerobic digestion process. In this case, AOP technologies can be effectively used, as proven by our previous studies [9]. The experimental data obtained from this work and previous studies [8,17,31] were summarized, and models of the effects of anaerobic conversion electrolysis were developed, taking into account the influencing factors on the condition of the activated sludge (Figure 10).

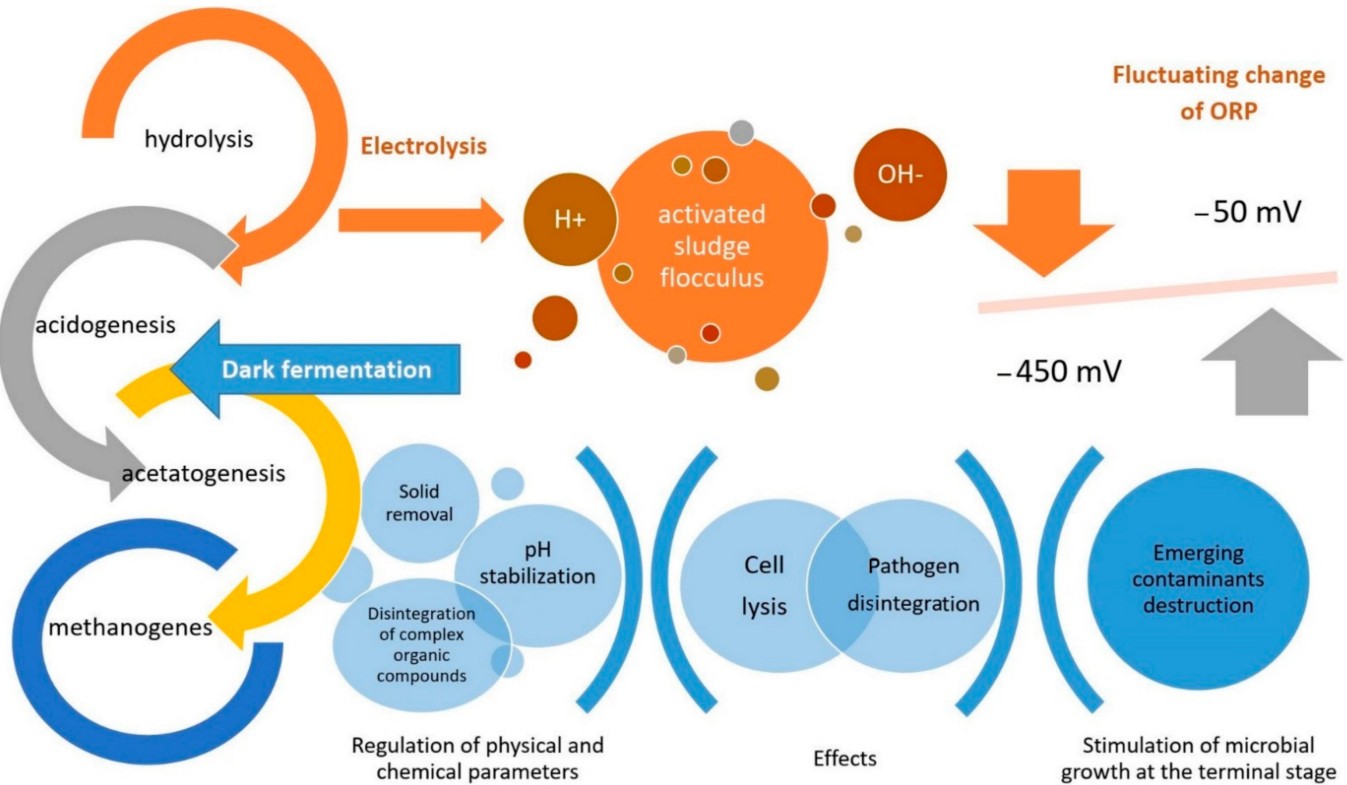

**Figure 10.** Summarized model of the effects of electrolysis on the process of anaerobic digestion in MECs, color identification: orange—hydrolysis stage with electric current treatment of substrate during anaerobic digestion with introduction of hydrogen ions and changes in ORP; gray—acidogenic stage of anaerobic digestion; yellow—acetatogenic stage of anaerobic digestion; dark blue—methanogenic stage of anaerobic digestion; light blue—main effects of electrolysis treatment on dark fermentation and anaerobic digestion in general; shades of brown—gas phase.

Thus, a model was formed based on both this experiment and previous studies as a generalized formalization of the effects of electrolysis on different types of substrates and the metabolic activation of microorganisms during anaerobic digestion.

## 4. Conclusions

The structure of anaerobic sludge changes in comparison with initially activated sludge; under the influence of an electric current toxic or nondegradable substances were oxidized to be biodegradable. $CO_2$ and $CH_4$ yields increased compared to the conventional process. Additionally, in an MEC, the hydrogen sulfide concentration was initially more than 5000 ppm, as in conventional conditions. During this treatment, it fell on the 14th day to 487 ppm. Visually microscopying anaerobic activated sludge under the influence of electrolysis in MECs showed changes in the aggregation process itself, with the formation of cells of clusters of microorganism colonies with branches of a delineated shape. The ORP in this case has a fluctuating character, with a decrease to −450 mV and an increase to −50 mV. After the hydrolysis stage, there is a stabilization of the value in the low ORP levels, which corresponds to the state of strict anaerobiosis and is an indicator of the transition to the

next stage of anaerobic digestion. A model of the effects of electrolysis in MECs during the anaerobic conversion was developed, taking into account the influencing factors on the condition of the activated sludge.

Further research is needed to examine in detail the operating conditions of the combined electrolysis cell and anaerobic bioreactor system to ensure stable biogas production as well as assess the energy requirements associated with the configuration and periods of electro-oxidizing action and its parameters, such as the current strength and density at the hydrolysis stage and during the digestion process. It is also necessary to study the effect of electrolysis in the process of anaerobic digestion on the ammonia content in biogas. In this case, two directions of the implementation of MECs are possible: biohydrogen and biomethane production.

**Author Contributions:** Conceptualization and methodology, Y.C. and V.S.; writing—original draft preparation, experiment performing, and data analysis, Y.C. and V.C.; validation, writing—review and editing, and funding acquisition, M.B.; formal analysis and technical support, V.C.; resources and writing—review, N.J. All authors have read and agreed to the published version of the manuscript.

**Funding:** This research was funded by VEGA—Scientific Grant Agency of the Ministry of Education, Science, Research and Sport of the Slovak Republic and the Slovak Academy of Sciences, grant number 1/0419/19.

**Institutional Review Board Statement:** Not applicable.

**Informed Consent Statement:** Not applicable.

**Data Availability Statement:** Not applicable.

**Acknowledgments:** We are thankful for the support provided by the International Innovation and Applied Center "Aquatic Artery" (Sumy, Ukraine), which allowed this scientific cooperation to start, and to the Czech government support provided by the Ministry of Foreign Affairs of the Czech Republic, which allowed this scientific cooperation to start within the project "AgriSciences Platform for Scientific Enhancement of HEIs in Ukraine". This research project was carried out as planned research projects of the Department of Ecology and Environmental Protection Technologies of Sumy State University, related to the topic "Assessment of the technogenic load of the region with changes in industrial infrastructure" according to the scientific and technical program of the Ministry of Education and Science of Ukraine (state registration No 0121U114478); joint Ukrainian–Czech R&D project "Bioenergy innovations in waste recycling and natural resource management", 2021–2022.

**Conflicts of Interest:** The authors declare no conflict of interest.

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
