# Peer review of "Effect of Electrolysis on Activated Sludge during the Hydrolysis and Acidogenesis Stages in the Anaerobic Digestion of Poultry Manure"

_sustainability, doi:10.3390/su14116826_

Round 1

Reviewer 1 Report

The topic is very interesting for AD, however the manuscript must be rewritten to improve the quality.

It is difficult to understand the real purpose of the study. For example, poultry litter is mentioneted in  the tittle but in the introduction there is just a small mention (line 118 - 122)

MM section is confuse, is it a co-digestion process?

Results and discussion are poorly presented and discussed.

Author Response

Thank you very much for your review of our manuscript. Answers to questions and comments are enclosed in the attached file.

Reviewer 2 Report

In the text, when referring to the figure, it should be written Figure, not Fig.

No reference is made to Figure 3 in the text, although it is represented.

When speaking in the text about fig. 6 a and 7 a probably is 5 a and 6 a.

Figure 7 b is discussed in the text (line 271) but is not shown.

No reference is made in the text to Table 1.

Figure 1 based on what was obtained? the graphs are represented from 2006 (a) respectively 2010 (b) although in the bibliography the resources indicate from 2013.

At the same time, the bibliographic sources on the basis of which fig 1 a and b were made do not appear.

Bibliographic sources are very few in number.

The sample for SEM  was dry, but how was it prepared? How was it read at SEM, with what degree of magnification? What type of SEM was used?

The References chapter does not meet the requirements of the journal.

Author Response

Thank you very much for your review of our manuscript. Responses to questions and comments are enclosed in the attached file.

Reviewer 3 Report

Comments on the manuscript “Effect of electrolysis on activated sludge during hydrolysis and acidogenesis stages in the anaerobic digestion of poultry manure” by Yelizaveta Chernysh, Magdalena Balintova, Vladimir Shtepa, Viktoriia Chubur and Natalia Junakova submitted to Sustainability MDPI.

The problem of managing of organic waste, is very crucial in terms of time and energy consumption. Of importance is the production of energy by anaerobic digestion. Quite often, due to the difficulty of primary decomposition of organic substances, this process is hindered and therefore all kinds of pre-treatment are used.

In the presented publication, the authors have attempted electrolytic activation of poultry manure in microbial electrolysis cell. They used the conventional method and additionally introduced electrodes into the system.

The authors claim to have developed a model of the effects of electrolysis during anaerobic digestion. In the experiment they used only one type of substrate and one, constant distance between the electrodes. Meanwhile, to talk about the model should be used more combinations of these factors.  There are many similar solutions in the literature, but the authors focused on a cursory search of Scopus and WoS databases. The only novelty not yet published in this context is the use of quail manure

Lines 45-47 are neither connected to the proceding sentences nor to the following ones. Additionally, Fig 1 does not show actual data. The works on application of electrolysis in biogas production is not son rare as Authors state. Authors wrote that they made a research based on words “biogas” and “electrolysis”. It looks that in the case of Scopus and WoSc it is not accurate as search on google scholar made without citations shows 16400 hits in the years 2010-2022, of which 1910 publications found are from 2022. The only true is that the interest in this area is increasing.

So my suggestion is not to show the numbers or figures like Fig 1 but really to make a research in databases concerning the subject of electrolysis treatment in for biogas production.

The introduction should be expanded to include relevant findings in this matter.

Materials and Methods

The information is given in the wrong order. First the characteristics of the manure parameters appear (Table 1) and only then the authors state how they were measured.

First of all the information about the source of quail manure should be given……..line 133….quail source originated from/ was a gift from/ specify name of the breeder/ producer/ city/ country/ obtained amount.

It should be followed by information what parameters were considered and how they were measured.

Line 138 tap water-please give chemical characteristics, at least pH and  chlorides.

Line 160 how was hydrogen content measured? It should be given here

Line 169 what stabilization of inoculum means?

Table 2 Does FS mean Full Scale? Add it as a footnote

Line 253 the reaction equation should read:

 ??2+4?2→??4+2?2?   (do not forget the correct reaction factors)

Results and Discussion, Conclusions

Please study the literature, compare and refer accordingly

References

Please change the layout of the references according to

Language

The language of the manuscript needs much revision. Please check the entire manuscript with particular attention to the lines

Other remarks

Decide whether to use pretreatment or pre-treatment? Check throughout the text

lines 6-12, align left as in other lines (lines 6-12)

line 54 biogas can not be divided as bi-ogas

Line 152 what thermo-plastic means?  thermoplastics include acrylic, polyester, polypropylene, polystyrene, nylon, Teflon etc……please specify

Author Response

(The authors gave the same response as above.)

Round 2

Reviewer 1 Report

The manuscript qualitiy was improved.

Reviewer 3 Report

The manuscript has been thoroughly improved. Please check the minor issues with language.